# Aesthetic Gingival Melanin Pigmentation Treatment in Smokers and Non-Smokers: A Comparison Study Using Nd:YAG Laser and Ceramic Bur

**DOI:** 10.3390/jpm13071034

**Published:** 2023-06-23

**Authors:** Massa Mahayni, Omar Kujan, Omar Hamadah

**Affiliations:** 1Department of Oral Medicine, Faculty of Dental Medicine, Damascus University, Damascus P.O. Box 30621, Syria; massa.mahayni@gmail.com (M.M.); omar.hamadah@damascusuniversity.edu.sy (O.H.); 2UWA Dental School, The University of Western Australia, 17 Monash Avenue, Nedlands, WA 6009, Australia

**Keywords:** aesthetic, gingival melanin pigmentation, Nd:YAG laser, ceramic bur, smokers, non-smokers

## Abstract

Aesthetic concerns are increasing rapidly; thus, several approaches have been suggested for treating gingival melanin pigmentation. Lasers have been reported as an effective new tool, and the Nd:YAG laser beam has an affinity for melanin and haemoglobin. However, ceramic gingival bur is simple and has less bleeding effect during operation than conventional techniques. This study aimed to compare the outcomes of gingival depigmentation using the Nd:YAG laser and ceramic bur in two different groups (smokers and non-smokers). A total of 40 patients presenting with gingival melanin pigmentation were enrolled in this split-mouth study. The sample was divided into two groups: smokers and non-smokers. Treatment was performed using the Nd:YAG laser (3 W, 60 mJ/pulse, and 50 Hz) and ceramic bur with a one-week interval between the two methods. Clinical indices were recorded, including intraoperative bleeding, wound healing, post-operative pain, and the recurrence of pigmentation, and follow-up periods were determined in the 3rd, 6th, and 9th months postoperatively. Both treatments promoted a similar pain experience and recurrence rate of pigmentation (*p* > 0.489, *p* = 1.000, respectively). Bleeding during surgery and complete healing recovery after one week were statistically significantly higher when using ceramic bur (*p* = 0.00, *p* = 0.041, respectively). Concerning the effect of smoking on the treatment, a higher recurrence rate was observed in SG than N-SG in laser sites (50%, and 95%, respectively) and bur sites (60%, and 85%, respectively), but statistically no significant difference was observed (*p* > 0.080). In conclusion, both procedures are adequate for aesthetic gingival depigmentation treatment. The Nd:YAG laser showed greater effectiveness in controlling bleeding, while ceramic bur showed a faster clinical recovery. Furthermore, smokers were more likely to have low depigmentation treatment stability.

## 1. Introduction

Healthy mouth components play a significant role towards defining a confident and attractive smile. An essential element of smile harmony is the colour of the gingival tissue. Gingival pigmentation is an aesthetic issue, especially for individuals with excessive exposure. The colour and clinical appearance of oral pigmentation can vary from brown to black, depending on the melanin production amount and the depth of the pigment position within the mucosa [1,2]. Melanocytes reside within the oral epithelium basal layer [1]. The melanocytes synthesise melanin, which is then transmitted via melanosomes to adjacent keratinocytes [3]. Clinically, it seems to be a bilateral, well-defined, ribbon-like strip that typically spares the marginal gingiva [4]. The attached gingiva is the most prevalent location [4]. Commonly, it is seen in black skin, including Asians and Mediterranean population individuals. However, it is not directly linked to skin colour [2]. The pigmentation of the oral mucosa includes a group of different entities, ranging from physiologic melanin pigmentation to melanoma [2]; physiologic pigmentation and smoker’s melanosis are mainly the most common [5].

Physiologic pigmentation is caused by the increased activity and number of melanocytes, and appears in the first two decades of a patient’s life but may not be discovered until later [6]. A smoker’s melanosis affects the maxillary and mandibular anterior aspects of the gingiva [7]; the labial gingiva is the most common location of a smoker’s melanosis. Furthermore, any oral region can be affected [1]. Smokers’ melanosis present as diffuse, patchy, irregular pigmentations [8]. Women are mainly affected, due to the role of the estrogen hormone. The histology features of smokers’ melanosis are like those described for physiological pigmentation [1]. Even in individuals where the genetic factors stimulate melanocytes, these cells were observed to have the capacity to increase their melanin production when activated by smoking [9]. Although the pathogenetic reasons are still not recognized, melanin may protect the mucosa from the thermal smoke effect or the cigarette irritant impact [8]. This pigmentation is caused by the activation of melanocytes due to polycyclic amines (nicotine and benzopyrenes), which increase melanin production to protect oral mucosa against tobacco agents [10]. It is well-recognised that smokers show a higher risk of gingival melanin pigmentation than non-smokers, and smoking cessation results in its reduction [11].

Various techniques have been introduced to the aesthetic removal of gingival pigmentations, such as surgical scraping, abrasion using diamond bur or ceramic bur [12,13], electrosurgery [14], cryosurgery [15], chemicals [16], and lasers [17]. Burs have been used widely in terms of depigmentation. This technique was reported as effective, non-invasive, and acceptable to patients and operators. Furthermore, it is a cost-effective technique and does not require specific instruments [13,18,19]. Compared with other abrasion techniques, ceramic bur showed the additional feature of minimal bleeding. The rotational energy provides thermal coagulation by sealing vessels [13,20]. Thus, no dressing pack is needed. Lately, laser ablation has been suggested as one of the most suitable, effective, and convenient methods for gingival depigmentation [17]. Several types of lasers have been used for the aesthetic treatment of gingival depigmentation, including diodes [21], Nd:YAG [22], Er:YAG [23], Er,Cr:YSGG [24], and CO_2_ [25] lasers. The absorption spectrum of melanin pigment ranges between 351 and 1064 nm, and the ability to absorb the laser beam depends on the laser wavelength and its ability to penetrate tissue [18]. Compared to other wavelengths, the Nd:YAG laser (1064 nm wavelength) has a particular beam for melanin and haemoglobin [26].

Hence, this study aimed to compare the outcomes of gingival depigmentation using the Nd:YAG laser and ceramic gingival trimming bur in two different groups (smokers and non-smokers) to find out the most appropriate technique for patients and clinical applications and to report on how smoking affects the outcome measures of comparisons. Results scales of comparisons include intraoperative bleeding, wound healing, post-operative pain, and recurrence in up to 9 months of follow-up sessions.

## 2. Materials and Methods

### 2.1. Study Design

This study was an experimental clinical trial with a split-mouth design comparing Nd:YAG laser and ceramic bur techniques for the aesthetic treatment of gingival melanin hyperpigmentation. All procedures were performed by the same operator at the Laser Department of the Faculty of Dental Medicine, Damascus University, between July 2018 and December 2021. The Local Ethics Research Committee approved the study at Damascus University (No. 2582-SM). All patients were informed with a detailed description of the proposed treatment and signed an informed consent form.

A total of 40 patients (22 males and 18 females, aged 18 to 30 years) presenting with gingival pigmentation were enrolled in this study.

### 2.2. Inclusion and Exclusion Criteria

Inclusion criteria included patients presenting with diffuse, bilateral, and symmetric gingival hyperpigmentation in the buccal maxillary and mandible gingiva. The participants were randomly divided into two groups, the first group presented with physiologic hyperpigmentation and the second group presented with smoker’s melanosis. Both groups were classified as (class 3) or (class 4) according to Hedin classification [27], which describes the extent of the pigmented area, and as either moderate (class 2) or severe (class 3) according to the Dummet classification (DOPI) [28], which describes the intensity of the pigmentation. The exclusion criteria were: (1) systemic diseases or hormonal disorders associated with gingival pigmentation, (2) systemic conditions that could affect healing and coagulation (diabetes, autoimmune disease, and leukaemia), (3) drugs intake, mainly those associated with gingival pigmentation (antimalarials, tetracycline, minocycline, ketoconazole), (4) periodontal diseases, and (5) pregnancy and lactation.

### 2.3. Clinical Indices

All patients were classified before treatment and during the subsequent follow-ups (at the 3rd, 6th, and 9th month post-OP) depending on the intensity and extent of gingival pigments according to the following indices by two independent clinicians:

Dummet Oral Pigmentation Index (DOPI) [28]: (0) pink tissue, no clinical pigmentation, (1) mild light brown tissue, mild clinical pigmentation, (2) medium brown or mixed brown and pink tissue, moderate clinical pigmentation, (3) deep brown/blue–black tissue, heavy clinical pigmentation

Hedin melanin index [27]: (0) no pigmentation, (1) one or two solitary units of pigmentation in the papillary gingiva, (2) >3 units of pigmentation in the papillary gingiva without formation of a continuous ribbon, (3) one short continuous ribbon of pigmentation, (4) one continuous ribbon, including the entire area between the canines.

Bleeding index [22]: The assessment was performed using visual examination and based on the amount of bleeding encountered during the procedure and the ease of performing the procedure. It was assessed on the following scale: 0 = no bleeding, complete homeostasis, 1 = isolated bleeding points during surgery, 2 = mild bleeding, but clear field, 3 = moderate/severe bleeding, difficulty in procedure.

Evaluation of pain after operation VAS: A visual analogue scale (VAS) was used to measure pain on the same day, 3rd, and 7th day. The VAS consists of a horizontal line 100 mm long, marked at the left end as (no pain) and at the right end as (maximum pain). The patient placed a mark on the scale that coincided with the pain level.

Evaluation of gingival healing: Healing was evaluated depending on the clinical gingival appearance on the 7th day and after 1 month post OP using the following wound healing index: A: complete epithelialisation, B: incomplete or partial epithelialisation, C: ulcer, D: tissue defect or necrosis.

Photos were taken using a Nikon D3300 (Nikon, Melville, NY, USA) with a Nikon AF-S DX NIKKOR 18–55 mm 1:3.5-5.6G VR II Micro lens.

### 2.4. Treatment Protocol

Each patient received both treatments separately, with an interval of one week. On one randomly chosen side, the Nd: YAG laser device (PLUSMASTER 600 IQ; American Dental Technologies, Corpus Christi, TX, USA, September 2002) was used with a free running pulse mode with the following parameters: wavelength 1064 nm, power 3 W, energy 60 mJ, and frequency 50 Hz. A 320 nm fibre diameter handpiece was used in a contact mode (Figure 1A). Conversely, the gingival trimming ceramic bur (Ceratip) which was made of a mixed ceramic (zircondioxide oxide partly stabilized by yttrium and aluminium ceramic), was operated in the high-speed mode without cooling to preserve the thermal coagulation action, causing blood vessels to seal for minimal bleeding (Figure 1B and Figure 2A). For both treatments, 2% lidocaine with 1:80,000 epinephrine local anaesthesia was infiltrated.

### 2.5. Nd:YAG (1064 nm) Laser Procedure

For safety measures, protective goggles were worn by staff and patients. The lasing fibre tip was stripped by 2 mm and initiated properly with an articulating paper. In the beginning, the pigmented area margins were selected, as shown in (Figure 2B). The handpiece was angled at an external bevel of 45° and moved in quick, gentle, and short-range strokes from the anterior to the posterior region with the cervico-apical direction of each pigmented area (Figure 2C). Care was taken to avoid damage to the adjacent tooth surfaces and surrounding tissues during ablation. The operated area was webbed with wet sterile gauze to remove carbonised debris. Gingiva was irradiated until the complete removal of pigmentation.

### 2.6. Ceramic Bur (Ceratip) Procedure

The ceramic bur was used at 300,000–450,000 RPM without cooling, as recommended by the manufacturer. There, the bur generates enough heat during rotation to preserve thermal tissue coagulation, allowing gingival trimming with minimal bleeding. The tip was applied parallel with the tissue and feather-light brushing strokes with gentle pressure were applied intermittently without holding or moving the bur forwards or backwards in the same place (Figure 2D). Caution was taken to ensure that all remnants of the pigmented layer were clinically removed.

The tissue removal for both laser and bur was completed by maintaining a distance of nearly 1 mm from the gingival margin to avoid the risk of gingival recession as much as possible (Figure 2E,F).

### 2.7. Post-Operative Instructions

No periodontal pack was applied to support the healing process. Written postoperative instructions were given to the patients. The patient was informed to follow the oral hygiene instructions and cautioned to avoid brushing the treated sites on the day of the treatment, and to avoid mechanical trauma and acidic beverages during the first week.

### 2.8. Statistical Analysis

The sample size was calculated to be 44 patients (22 in each group), using G* Power 3.1.9.4 (G*Power, Düsseldorf, Germany) and considering an alpha of 0.05, power of 80%, and 1.10 as effect size (depending on the visual analogue scale (VAS) values given in a previous paper [22]). Four patients’ data were incomplete, and consequently the sample size was decreased to become 40 patients (20 in each group).

The data were entered into Microsoft Excel and analysed using the Statistical Package for Social Sciences (SPSS) version 22.0 (IBM, Armonk, NY, USA). A Kolmogorov–Smirnov test was used to check the normality of the data, and they were not normally distributed. Hence, suitable nonparametric tests (Mann–Whitney U test) were used to test the significance level between the two study groups. Differences between groups and time points were compared using the Wilcoxon Signed Rank test. For the analysis, *p* < 0.05 was considered statistically significant.

## 3. Results

Intraoperative bleeding index. Bleeding during the treatment for both procedures is shown in Table 1 and Table 2 and Figure 3 and Figure 4. Results showed that 90% of regions treated with Nd:YAG groups scored grade 0 on the bleeding index (85% of SG and 95% of non-SG), while only 10% scored grade 1 (15% of SG and 5% of N-SG). On the other hand, 7.5% of regions treated with the bur group scored grade 0 (7.5% of SG and 0% of N-SG), 55% scored grade 1 (55% of SG and N-SG), 22.5% scored grade 2 (30% of SG, and 15% of N-SG), and 15% scored grade 3 (30% of SG). Bleeding in the bur group (mean rank = 57.75) was statistically significantly higher than in the Nd:YAG group (mean rank = 23.25) (*p* = 0.00). At the same time, the difference was insignificant between SG and N-SG patients in terms of bleeding for each treatment (*p* > 0.054).

VAS Index of Pain Measurement. The VAS index of pain is shown in Table 3 and Table 4 and Figure 5 and Figure 6. On the same day, and at the 3rd and 7th days post-OP, relating to pain score, the majority of patients treated with Nd:YAG and bur treatment had a painless experience on the same day of the procedure (82.5%, and 92.5%, respectively) compared with 22.5% and 20%, respectively, at the 3rd day, and 0% for both groups after the 7th day. There was no significant difference between treatment groups on the same day, 3rd, and 7th days (*p* >0.489). On the other hand, 75% of SG and 90% of N-SG in Nd:YAG treated regions, and 85% of SG and 100% of N-SG in the bur group had a painless experience on the same day of the procedure. On the 3rd day post-OP the figures were 5% of SG, 40% of N-SG, and 10% of SG and 30% of N-SG, respectively. However, no significant differences were observed between S and N-S groups on the same day and 7th day (*p* > 0.713).

Healing index. Healing index results are shown in Table 5 and Table 6 and Figure 7 and Figure 8. The current study analysis exposed that more than two-thirds in all groups had complete healing seven days following the treatment; Nd:YAG-treated regions scored 0 in 67.5% of areas (70% of SG, and 65% of N-SG), while the bur-treated group scored 0 in 87.5% of regions (90% of SG, and 85% of N-SG). A significant difference was observed between the Nd:YAG group (mean rank = 44.3) and the bur group (mean rank = 36.66) (*p* = 0.041), but there was no significant difference between SG and N-SG in each treatment (*p* > 0.689). After 1 month, all patients scored 0 on the healing index in both groups, with no significant difference (*p* = 1.000).

Recurrence according to Hedin and Dummet indices. Each treatment was able to achieve the removal of gingival melanin pigmentation perfectly. Hedin and Dummet indices showed 0 degrees in all groups after two weeks of treatment. The results regarding the gingival pigmentation indices, Hedin and Dummet, are presented in Table 7 and Table 8 and Figure 9 and Figure 10. For the Nd:YAG-treated-regions group, recurrence was observed at three months in 17.5% of regions (25% of SG and 10% of N-SG). At six months, the recurrence increased and was observed in 42.5% of regions (50% of SG and 35% of N-SG), and at nine months recurrence was observed in 12.5% of regions (20% of SG and 5% of N-SG). For the ceramic-bur-treated-regions group, recurrence was observed after three months in 17.5% of regions (20% of SG and 15% of N-SG); after six months, recurrence increased and was observed in 40% of regions (45% of SG, and 35% of N-SG), and after nine months recurrence was observed in 15% of regions (20% of SG, and 10% of N-SG). No significant differences between either group were realised in the 3rd, 6th, and 9th months (*p* = 1.000). Also, the difference was not significant between SG and N-SG patients during all follow-up sessions for each treatment (*p* > 0.08) (Figure 11 and Figure 12).

## 4. Discussion

Increasing aesthetic concerns are motivating patients to seek cosmetic treatment for their unwanted pigmented gingiva, especially if the gingival pigmentation is visible during talking or smiling. 

Several approaches have been suggested for removing gingival pigmentations; the chosen method was based mainly on personal preference and clinical practice experience. Subsequently, lasers have been reported as an effective new tool due to their distinct advantages, such as coagulation [29] and pain reduction [17]. The Nd:YAG laser has a particular feature when pigments exist; its beam is affixed to melanin and haemoglobin [26]. Thus, it works more efficiently when used for gingival pigmentation treatment. In addition, different parameter settings have been used [17]. Moreover, the Nd:YAG laser has more penetration depth than other lasers, whereas the unfavourable heat produced by Nd:YAG is less than the diodes. Thus, we used the lowest energy levels (3 watts) to reduce the unfavourable heat.

The Nd:YAG laser was preferred for use as an alternative tool to conventional procedures [22,30]; using such methods resulted in unpleasant bleeding during and after the procedure [31], and the need for the periodontal pack to cover the treatment area, as well [32]. However, ceramic bur has an additional feature of less bleeding effect during operation than conventional techniques. As a result, vessels are sealed by the rotational thermal effect [13], and therefore there is no need for any dressing or periodontal pack after surgery. In addition, this technique is simple and does not require any advanced equipment.

However, up to now, no study has compared the use of the Nd:YAG laser and ceramic bur for treating GMP. Moreover, only a few studies have compared the treatment of GMP between smokers and nonsmokers. Thus, our study compared the clinical results when using the Nd:YAG laser and ceramic bur for aesthetic gingival melanin depigmentation. Also, smokers and non-smokers were compared in each treatment group.

In the current study, DOPI was used for the intensity and the Hedin index was used for the extent of pigmentation. The DOPI and Hedin’s index scored 0 after 2 weeks PO compared with the baseline in Nd:YAG- and bur-treated sites. Nd:YAG lasers and ceramic bur were both effective for treating GMP.

Gingival depigmentation stability through time is a significant factor that should be taken into account when evaluating the success of GMP treatment. The recurrence rate depends on racial factors, the ability to remove melanin and melanocytes, smoking habits, and the laser’s power [17]. Moreover, the recurrence rate may also vary depending on the treatment modalities and the duration of follow-up [19].

In this study, recurrence happened in 72% of treated sites after nine months of follow-up. Comparing the two methods, almost the same recurrence results were shown in both laser and bur at 3, 6, and 9 months. Recurrence was more likely to happen at six months, with a significant difference from results at 3 and 9 months. However, the recurrence conditions were much less than the pigmentation conditions at baseline in terms of extent and intensity. We expect that recurrence happened because some melanocytes had been left behind at marginal or adjacent gingival areas, or the high activity of these melanocytes may have resulted in the pigmentation recurrence.

Our results are similar to Negi et al. [13]’s study, which compared a ceramic bur technique with a diode laser in a split-mouth design. They noticed no statistically significant differences between the two methods at six months, and recurrence occurred in 16 of 20 patients. Basha et al. [22] observed recurrence in 65% of patients treated with Nd:YAG laser sites. The cases were followed for six months, and the pigmentations that appeared were in the form of small dots or streaks in the interdental areas or on the attached gingiva. Alasbahi et al. reported recurrence in 35% of cases treated with the Nd:YAG laser. The treatment was performed on 20 patients with racial GMP for a follow-up period of up to 9 months. Ribeiro et al. [19] showed that 45% had a recurrence after six months of follow-up after using the Nd:YAG laser.

Conversely, Atsawasuwan et al. [30]’s study reported no recurrence using an Nd:YAG laser up to 12 months. The procedure was performed on four patients. Armojida et al. [33]’s study reported no cases of recurrence within at least one year of follow-up when using a Q-Switched Nd:YAG laser, and this was justified by the ability to apply the treatment on the gingival margin and interdental papilla without taking caution for the risk of a gingival recession when using a Q-switched Nd:YAG laser.

Concerning the effect of smoking on gingival pigmentation, this study showed a higher recurrence rate in SG than N-SG, but statistically no significant difference was observed. Nammour et al. [11] showed that recurrence happened for both smokers’ and non-smokers’ groups; they observed that a diode laser had the most extended period before recurrence compared to patients treated with CO_2_ and Er:YAG. Moreover, they showed that non-smokers had the longest time before recurrence compared to smokers.

In the current study, the bur-treated sites healed faster than laser-treated sites, with statistically significant differences between the groups (*p* = 0.041). As Nd:YAG has a more thermal impact on the tissue, the thermal effect can delay the healing process, and it is responsible for these results; however, it should be mentioned that, after two weeks, all the cases in all the groups were healed completely. In the findings from Negi et al. [13]’s study, complete wound healing was observed in more patients on the 7th day on the bur-treated sites compared to the laser-treated sites. Mani et al. reported that wound healing after using a diode laser requires more time than bur and conventional surgical techniques [34]. Alasbahi et al. also observed incomplete healing in 60% of Nd:YAG-treated sites after seven days. Romanos et al. [35]’s study reported the delayed healing of the Nd:YAG laser treatment compared to scalpel incisions when using 3 W of power and a 20 Hz pulse rate. Faster healing of soft tissue wounds after laser treatment is wavelength specific and highly critical to energy density [26]. At the same time, the difference between SG and N-SG for each treatment was not significant.

Clinically, the bleeding index results showed less bleeding while using the Nd:YAG laser than with the ceramic bur, and the difference was statistically significant (*p* = 0.00). As we mentioned before, it is justified that Nd:YAG had deeper penetration, causing a thermal impact on the tissue and providing more coagulation efficacy. Identical results were shown in multiple previous studies. The observations made by Negi et al. [13] showed that intra-operative moderate bleeding was seen in 70% of the patients with a ceramic bur. In comparison, only 20% of the patients showed bleeding with a laser. Similarly, many studies noted that a high control of the bleeding was achieved when using Nd:YAG lasers [12,22,30].

According to the findings, a greater extent of pain was experienced on the side treated with the ceramic bur compared to the Nd:YAG laser procedure. Furthermore, no significant difference was observed between the groups. Ribeiro et al. [19] reported that a greater extent of discomfort/pain was experienced in the side treated with the scalpel technique compared to the Nd:YAG laser procedure. Negi et al. [13] found that pain perception was less in the diode laser group compared to the bur group. Pain reduction after laser application may be attributed to the formation of protein coagulating on the wound surface, acting as a biological dressing [36]. In addition, the heat generated during the rotation of the ceramic bur could seal the ends of sensory nerves.

The study might have a limitation of a relatively short follow-up period. In this study, patients were followed up at 9 months post OP, and a longer follow-up period would be eligible to determine the time required for the complete recurrence of GMP.

## 5. Conclusions

Both the Nd:YAG laser and ceramic bur techniques proved effective in achieving aesthetic satisfaction. Laser ablation is one of the most impressive techniques available presently. Conversely, the ceramic bur method is simple, non-aggressive, and inexpensive.

## Figures and Tables

**Figure 1 jpm-13-01034-f001:**
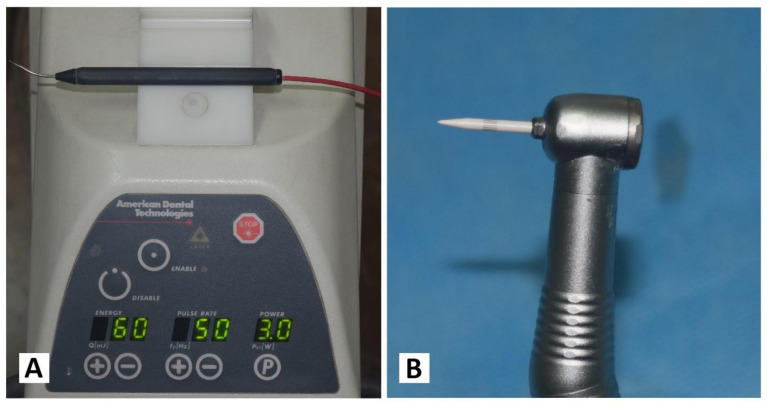
(**A**) Nd:YAG laser PLUSMASTER 600 IQ. (**B**) Gingival trimming ceramic bur (Ceratip).

**Figure 2 jpm-13-01034-f002:**
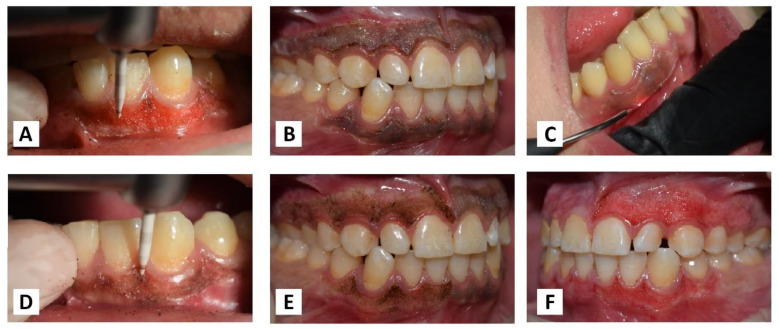
(**A**) Ceramic bur operated without cooling. (**B**) Selecting pigmented area margins. (**C**) Laser fibre angled at 45° in a contact mode. (**D**) Bur tip applied in parallel with the tissue. (**E**,**F**) Maintaining the distance of 1 mm from the gingival margin.

**Figure 3 jpm-13-01034-f003:**
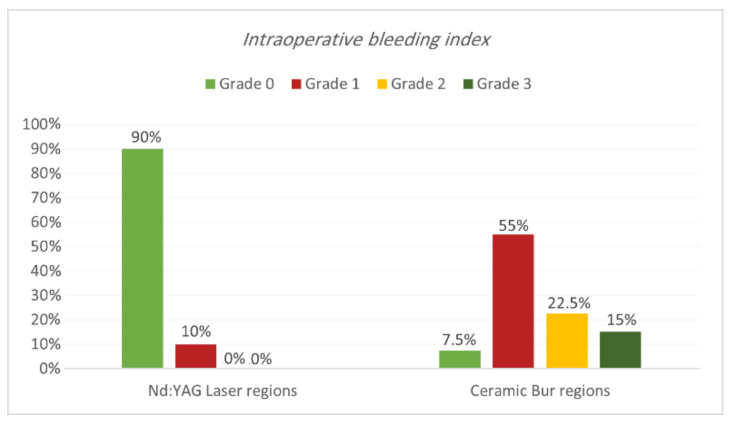
Comparison of the intraoperative bleeding index between the two treatment regions.

**Figure 4 jpm-13-01034-f004:**
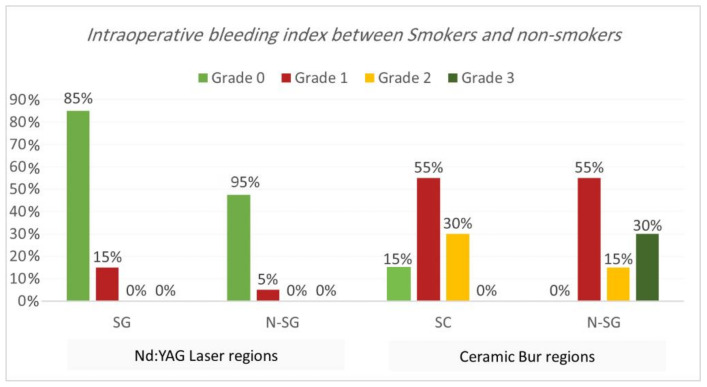
Comparison of the intraoperative bleeding index between smokers and non-smokers groups for each treatment.

**Figure 5 jpm-13-01034-f005:**
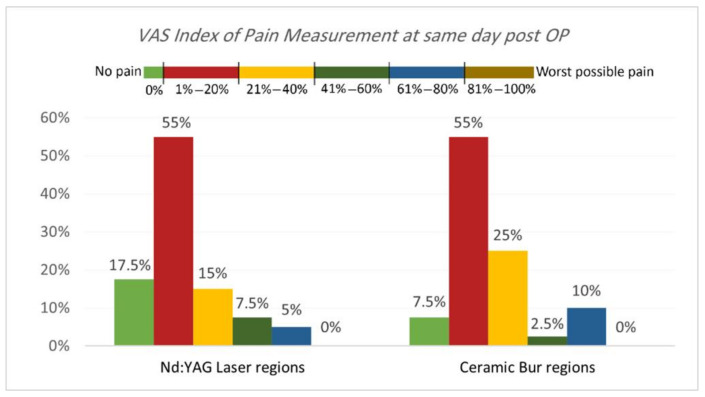
Comparison of the VAS Index of Pain Measurement between the two treatment regions on the same day post OP.

**Figure 6 jpm-13-01034-f006:**
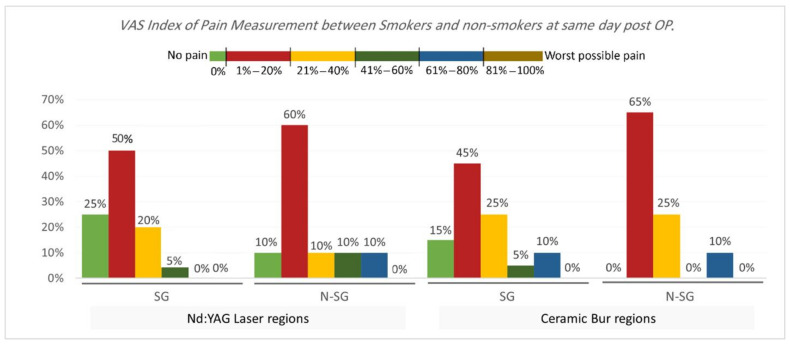
Comparison of the VAS Index of Pain Measurement between smokers and non-smokers groups for each treatment on the same day post OP.

**Figure 7 jpm-13-01034-f007:**
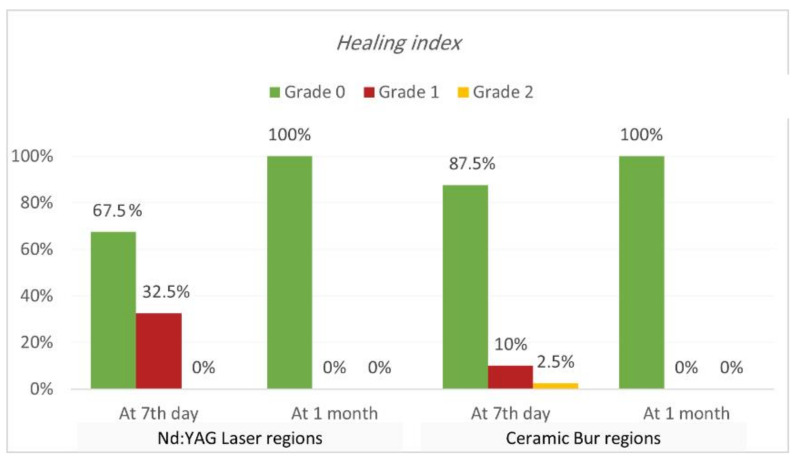
Comparison of healing index between the two treatment regions at various time intervals (at 7th day, and 1 month post OP).

**Figure 8 jpm-13-01034-f008:**
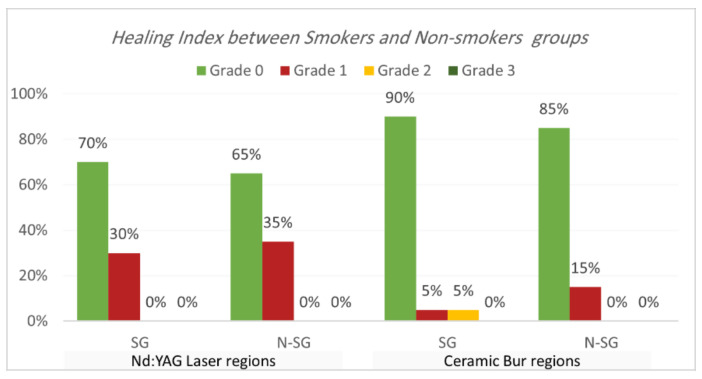
Comparison of healing index between smokers and non-smokers groups for each treatment after the 7th day post OP.

**Figure 9 jpm-13-01034-f009:**
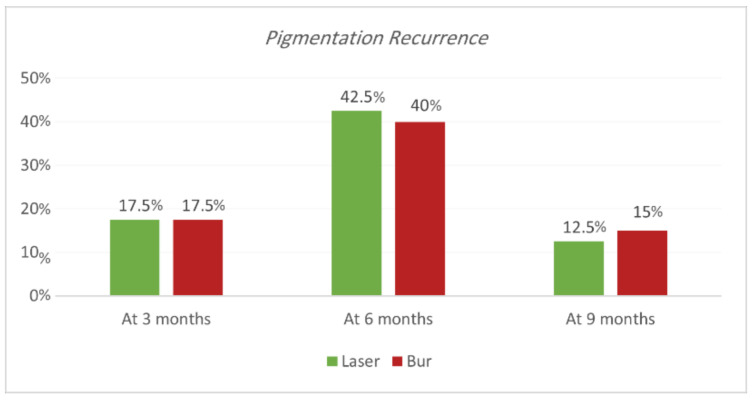
Comparison of recurrence rate according to Hedin and Dummet indices between the two procedures at the various time intervals.

**Figure 10 jpm-13-01034-f010:**
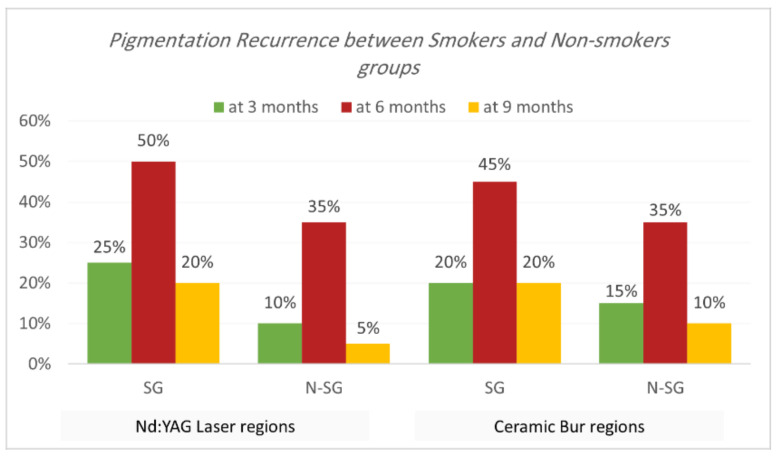
Comparison of recurrence rate according to Hedin and Dummet indices between smokers and non-smokers groups for each treatment at the various follow-up intervals.

**Figure 11 jpm-13-01034-f011:**
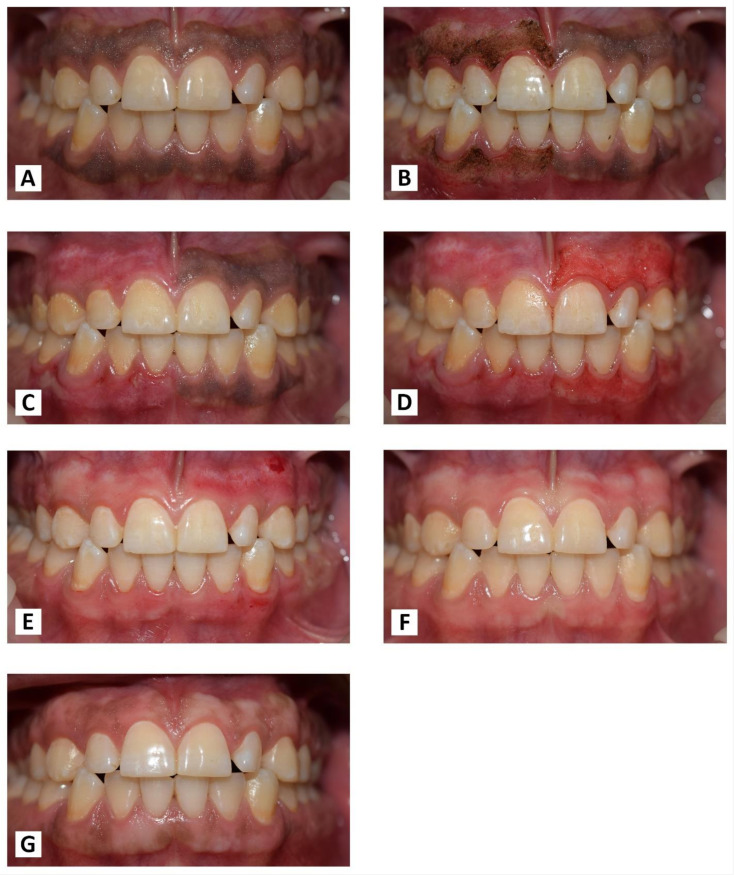
(**A**) Before treatment. (**B**) Immediately after laser treatment. (**C**) After one week of laser treatment. (**D**) Immediately after bur treatment. (**E**) After one week of bur treatment. (**F**) At one month post-OP. (**G**) At the nine-month follow-up period.

**Figure 12 jpm-13-01034-f012:**
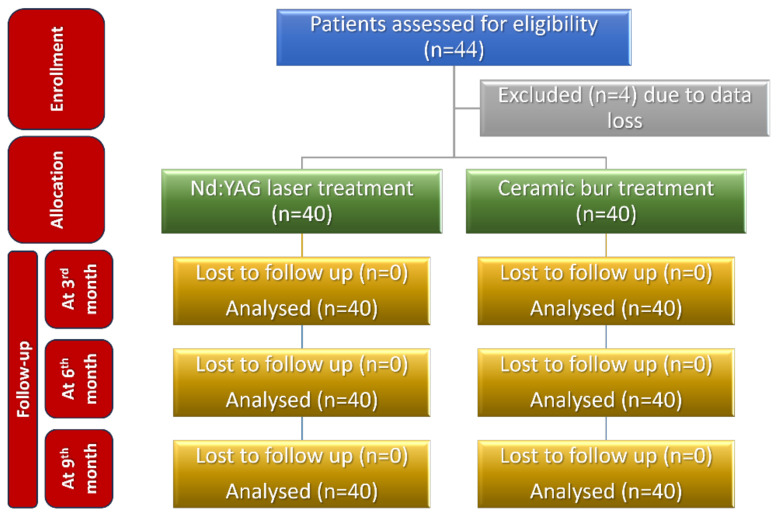
Consolidated Standards of Reporting Trials (CONSORT) flow diagram of patients’ recruitment, follow-up, and entry into data analysis.

**Table 1 jpm-13-01034-t001:** The comparison of the intraoperative bleeding index between the two treatment regions.

	**N**	**Mean Rank**	**Sum of Ranks**	Asymp. Sig. (2-tailed) 0.000
Nd:YAG Laser	40	23.25	930.00	
Ceramic Bur	40	57.75	2310.00	

**Table 2 jpm-13-01034-t002:** For each treatment, the intraoperative bleeding index between the smokers and non-smokers groups was compared.

	Nd:YAG Laser	Ceramic Bur
Mann–Whitney U	180.000	135.500
Wilcoxon W	390.000	345.500
Z value	−1.041	−1.928
Asymp. Sig. (2-tailed)	0.298	0.054

**Table 3 jpm-13-01034-t003:** The comparison of the VAS Index of Pain Measurement between the two treatment regions at the various follow-up intervals.

	Same Day	3rd Day Post OP	7th Day Post OP
Mann–Whitney U	728.500	798.000	800.000
Wilcoxon W	1548.500	1618.000	1620.000
Z value	−0.693	−0.027	0.000
Asymp. Sig. (2-tailed)	0.489	0.979	1.000

**Table 4 jpm-13-01034-t004:** The comparison of the VAS Index of Pain Measurement between smokers and non-smokers groups for each treatment at the various follow-up intervals.

	Nd:YAG Laser			Ceramic Bur		
	Same Day	3rd Day Post OP	7th Day Post OP	Same Day	3rd Day Post OP	7th Day Post OP
Mann–Whitney U	186.500	131.000	200.000	195.500	160.000	200.000
Wilcoxon W	396.500	341.000	410.000	405.500	370.000	410.000
Z value	−0.368	−2.554	0.000	−0.123	−1.549	0.000
Asymp. Sig. (2-tailed)	0.713	0.011	1.000	0.902	0.121	1.000

**Table 5 jpm-13-01034-t005:** The comparison of the healing index between the two treatment regions at the various follow-up intervals.

	7th Day Post OP			1 Month Post OP		
	N	Mean Rank	Sum of Ranks	N	Mean Rank	Sum of Ranks
Nd:YAG Laser	40	44.34	1773.50	40	40.50	1620.00
Ceramic Bur	40	36.66	1466.50	40	40.50	1620.00
Asymp. Sig. (2-tailed)	0.041			1.000		

**Table 6 jpm-13-01034-t006:** Comparing the healing index between smokers and non-smokers groups for each treatment at the various follow-up intervals.

	Nd:YAG Laser		Ceramic Bur	
	7th Day Post OP	1 Month Post OP	7th Day Post OP	1 Month Post OP
Mann–Whitney U	190.000	200.000	191.500	200.000
Wilcoxon W	400.000	410.000	401.500	410.000
Z value	−0.333	0.000	−0.401	0.000
Asymp. Sig. (2-tailed)	0.739	1.000	0.689	1.000

**Table 7 jpm-13-01034-t007:** The comparison of recurrence rates according to Hedin and Dummet indices between the two procedures at the various follow-up intervals.

	3rd Month Post OP	6th Month Post OP	9th Month Post OP
Mann–Whitney U	800.000	800.000	800.000
Wilcoxon W	1620.000	1620.000	1620.000
Z value	0.000	0.000	0.000
Asymp. Sig. (2-tailed)	1.000	1.000	1.000

**Table 8 jpm-13-01034-t008:** Comparing recurrence rates according to Hedin and Dummet indices between smokers and non-smokers groups for each treatment at the various follow-up intervals.

	Nd:YAG Laser			Ceramic Bur		
	3rd Month Post OP	6th Month Post OP	9th Month Post OP	3rd Month Post OP	6th Month Post OP	9th Month Post OP
Mann–Whitney U	170.000	170.000	160.000	190.000	180.000	180.000
Wilcoxon W	380.000	380.000	370.000	400.000	390.000	390.000
Z value	−1.233	−0.947	−1.749	−0.411	−0.637	−0.874
Asymp. Sig. (2-tailed)	0.218	0.343	0.080	0.681	0.524	0.382

## Data Availability

The data supporting this study’s findings are available on request from the corresponding author. The data are not publicly available due to privacy and ethical restrictions.

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
