# Peer review of "Aesthetic Gingival Melanin Pigmentation Treatment in Smokers and Non-Smokers: A Comparison Study Using Nd:YAG Laser and Ceramic Bur"

_jpm, 2023, doi:10.3390/jpm13071034_

Round 1
Reviewer 1 Report
This is very well written paper, I have few suggestions and corrections:
1. Were all treatments performed by the same clinician? If yes, state that in materials and methods, if no, add that to possible weakness of the study.
2. Have you performed power analysis to estimate the smallest sample size needed for an experiment? State that in “Statistical analysis” in materials and methods.
3. You should add “Statistical analysis” in materials and methods and describe used methods in this research.
4. Before conclusion, add paragraph with weaknesses of this study.
5. Page 5, 1st row: “Patients” should be with small letter “p”.
6. Page 18, 3rd paragraph: some references are missing (Negi et al., Basha et al., Alasbahi et al.).
Author Response
We thank the Reviewers for taking the time and effort to review the manuscript. We sincerely appreciate all your valuable comments and suggestions, which helped us improve our manuscript’s quality.
The manuscript has been revised per your suggestions, and here are below our point-to-point responses:
|
1. Were all treatments performed by the same clinician? If yes, state that in materials and methods, if no, add that to possible weakness of the study. |
Yes, all procedures were performed by the same operator. This note has been added to the mentioned spot. |
|
2. Have you performed power analysis to estimate the smallest sample size needed for an experiment? State that in “Statistical analysis” in materials and methods. |
The Sample size was calculated to be 44 patients (22 in each group) using G* Power 3.1. considering alpha of 0.05, power of 80%, and 1.10 as effect size (depending on the visual analog scale (VAS) values given in a previous paper *mentioned in the article*). Four patients’ data were lost; consequently, the sample size was decreased to 40 patients (20 in each group). |
|
3. You should add “Statistical analysis” in materials and methods and describe used methods in this research. |
1- “Statistical analysis” section has been added to the mentioned spot.
2- According to the second section of your question. - If you meant by the sentence *used methods* the description of Nd:YAG laser and ceramic bur. It has been described in the “2.4 Treatment protocol” section. Also, ceramic bur components were added to the above-mentioned section. - If you meant the Statistical test (ex: Wilcoxon, and Mann Whitney tests). It has been described in the “Statistical analysis” section, 2nd paragraph.
|
|
4. Before conclusion, add paragraph with weaknesses of this study. |
Two Limitations of this study have been added to the mentioned section. |
|
5. Page 5, 1st row: “Patients” should be with small letter “p”. |
Changes have been made as suggested. |
|
6. Page 18, 3rd paragraph: some references are missing (Negi et al., Basha et al., Alasbahi et al.). |
Thank you for the notice; the references have been added. |
Reviewer 2 Report
The paper titled "Esthetic Gingival Melanin Pigmentation Treatment in Smokers and Non-Smokers: A Comparison Study Using Nd:YAG Laser and Ceramic Bur" presents a comprehensive comparison between the use of Nd:YAG laser and ceramic bur for the treatment of gingival melanin pigmentation (GMP) in both smokers and non-smokers. The conclusions drawn from the study provide valuable insights into the effectiveness and outcomes of these treatment modalities.
The study highlights the increasing esthetic concerns of patients regarding unwanted pigmented gingiva, particularly when the pigmentation is visible during talking or smiling. It emphasizes the need for cosmetic treatments to address these concerns and explores various approaches to remove gingival pigmentation, with the selected method depending on personal preference and clinical experience.
The paper discusses the advantages of Nd:YAG laser in treating GMP, such as its affinity for melanin and hemoglobin, which allows for more efficient treatment. The laser's coagulation properties and pain reduction benefits are also highlighted. Additionally, the study acknowledges the higher penetration depth of Nd:YAG laser compared to other lasers while minimizing unfavorable heat production by using lower energy levels.
Comparatively, the ceramic bur technique is described as having the advantage of causing less bleeding during the operation. The rotational thermal effect of the bur seals vessels, eliminating the need for dressing or periodontal pack after surgery. This technique is considered simple and does not require advanced equipment.
The paper contributes to the existing literature by comparing the use of Nd:YAG laser and ceramic bur for GMP treatment and exploring the differences between smokers and non-smokers within each treatment group. The clinical results demonstrate the effectiveness of both Nd:YAG laser and ceramic bur in treating GMP.
Recurrence rate, an essential factor in evaluating the success of GMP treatment, is discussed in the paper. The study identifies several factors influencing recurrence, such as racial factors, the ability to remove melanin and melanocytes, smoking habits, laser power, and treatment modalities. The recurrence rate was found to be similar in both laser and bur treatment groups at different follow-up periods.
The study notes that healing was faster in bur-treated sites compared to laser-treated sites, attributed to the thermal impact of Nd:YAG laser that can delay the healing process. However, all cases in all groups achieved complete healing after two weeks. The bleeding index results indicate less bleeding with Nd:YAG laser compared to ceramic bur, while pain perception showed no significant difference between the two methods.
Overall, this paper provides valuable insights into the comparison of Nd:YAG laser and ceramic bur for esthetic gingival melanin depigmentation. The study's findings contribute to the understanding of treatment outcomes, recurrence rates, healing processes, and patient experiences. The conclusions drawn from the study highlight the effectiveness of both modalities in treating GMP and provide valuable guidance for clinicians in selecting the most suitable treatment approach based on patient preferences and clinical expertise.
Author Response
We want to take this opportunity to thank you for the effort and expertise that you contributed towards reviewing and summarizing this article.
Reviewer 3 Report
1. The tile all of "Figures" should be move to below the pistures.
2. The paragraph "The data were entered in Microsoft Excel and analysed using the Statistical Package for Social Sciences (SPSS) version 22.0. Kolmogorov–Smirnov test was used to check the normality of the data, and they were not normally distributed. Hence, suitable nonparametric tests (Mann– Whitney U test) were used to test the significance level between the two study groups. Differences between groups and time points were compared using the Wilcoxon Signed Rank test. For the analysis, P < 0.05 was considered statistically significant" move to the Methods part.
3. Did not see the data of smokers and nonsmokers in table 2, figure 4
4. Did not see the treatment regions in table 3
5. It is quite difficult to understnd the data in figure 5, 6, 7,8
6. Should have a diagram to describe the procedure of the experiments
it is good
Author Response
We thank the Reviewer for taking the time and effort to review the manuscript. We sincerely appreciate all your valuable comments and suggestions, which helped us improve our manuscript’s quality.
We revised the manuscript per your comments, and a response to the raised points is below.
|
1. The tile all of "Figures" should be move to below the pictures. |
Changes have been made as suggested. |
|
2. The paragraph "The data were entered in Microsoft Excel …." move to the Methods part. |
The mentioned paragraph has been moved to a newly created section, “Statistical analysis”, in the “Materials and Methods” part, as you requested. |
|
3. Did not see the data of smokers and nonsmokers in table 2, figure 4. |
According to: - Table 2: the given values in this table describe the differences between the smokers and non-smokers groups as a one-numbered given value (ex: P= 0.298). In other words, the column titled (Nd:YAG laser) has data representing the difference between the smoker’s and non-smoker’s groups, with similar data regarding the “Ceramic bur” column. - Figure 4: the data shown in this figure describes the smokers and non-smokers group results. Since “smokers group” and “non-smokers group” has been written as an abbreviation (SG, and N-SG) above “Nd:YAG” and “ceramic bur” boxes. |
|
4. Did not see the treatment regions in table 3 |
Similarly, as we described above, the given values in “Table 3” describe the differences between Nd:YAG treatment regions and ceramic bur treatment regions as a one-numbered given value. |
|
5. It is quite difficult to understand the data in figure 5, 6, 7,8 |
According to: - “Figures 5, 6” represent the results related to the VAS Index of Pain Measurement, some changes have been made to make the figures more clearly understood as you requested. The index’s ranges (written below) have been designed in a new way similar to the VAS scale. 0% no pain 1%-20% 21%-40% 41%-60% 61%-80% 81%-100% worst possible pain
- “Figures 7,8” which represent the results of the healing index regarding the two treatment regions on the 7th day and 1 month post-OP; some changes have been made.
Please let me know if this works and if you have any suggestions.
|
|
6. Should have a diagram to describe the procedure of the experiments. |
We appreciate your suggestion for having a diagram, a Consolidated Standards of Reporting Trials (CONSORT) flow diagram of patients' recruitment, follow-up, and entry into data analysis has been created as “Figure 12” |